# A Comprehensive and Integrative Approach to MeCP2 Disease Transcriptomics

**DOI:** 10.3390/ijms24065122

**Published:** 2023-03-07

**Authors:** Alexander J. Trostle, Lucian Li, Seon-Young Kim, Jiasheng Wang, Rami Al-Ouran, Hari Krishna Yalamanchili, Zhandong Liu, Ying-Wooi Wan

**Affiliations:** 1Jan and Dan Duncan Neurological Research Institute at Texas Children’s Hospital, Houston, TX 77030, USA; 2Department of Pediatrics, Baylor College of Medicine, Houston, TX 77030, USA; 3Department of Molecular and Human Genetics, Baylor College of Medicine, Howard Hughes Medical Institute, Houston, TX 77030, USA; 4Graduate Program in Quantitative and Computational Biosciences, Baylor College of Medicine, Houston, TX 77030, USA; 5USDA/ARS Children’s Nutrition Research Center, Department of Pediatrics, Baylor College of Medicine, Houston, TX 77030, USA

**Keywords:** MeCP2, data portal, Rett syndrome, MeCP2 duplication syndrome, RNA-seq, differential expression analysis, meta-analysis, mouse models

## Abstract

Mutations in MeCP2 result in a crippling neurological disease, but we lack a lucid picture of MeCP2′s molecular role. Individual transcriptomic studies yield inconsistent differentially expressed genes. To overcome these issues, we demonstrate a methodology to analyze all modern public data. We obtained relevant raw public transcriptomic data from GEO and ENA, then homogeneously processed it (QC, alignment to reference, differential expression analysis). We present a web portal to interactively access the mouse data, and we discovered a commonly perturbed core set of genes that transcends the limitations of any individual study. We then found functionally distinct, consistently up- and downregulated subsets within these genes and some bias to their location. We present this common core of genes as well as focused cores for up, down, cell fraction models, and some tissues. We observed enrichment for this mouse core in other species MeCP2 models and observed overlap with ASD models. By integrating and examining transcriptomic data at scale, we have uncovered the true picture of this dysregulation. The vast scale of these data enables us to analyze signal-to-noise, evaluate a molecular signature in an unbiased manner, and demonstrate a framework for future disease focused informatics work.

## 1. Introduction

Experimental reproducibility is a key issue in the life sciences. Big data integration and analyses can uncover valuable insights otherwise missed in individual studies [1], but curating, processing, and analyzing data at scale is challenging. Researchers could aggregate publicly available processed results, but doing so will inevitably yield inconsistencies between datasets. Handling a meaningful quantity of raw high-throughput data requires extensive time, experience, and computational resources, and it would be wasteful for every researcher to do this themselves.

Databases with abundant biological data exist but either do not focus on transcriptome perturbation (GTEx) [2] or require significant time and energy to extract and format data specific to a particular disease (ARCHS4) [3]. Similarly, many databases focus on specific model organisms and allow filtering by disease, such as Flybase [4] and the Rat Genome Database [5], but do not offer substantial disease-focused analysis. While there are some molecular-focused databases for well-studied diseases such as cancer (TCGA) [6], cBio Portal [7], and Alzheimer’s disease (AMP-AD) [8], there are no analogous databases for less-common diseases. The massive success of TCGA, in particular, makes the utility of disease databases clear.

Rett syndrome (RTT) is a severe neurodevelopmental disorder in girls caused by mutations in the X-linked gene for methyl-CpG binding protein 2 (*MECP2*). Development proceeds normally until 6–18 months, at which point it stalls and then regresses [9]. Symptoms and progression can vary substantially between individuals, and despite recent advances, we still do not completely understand MeCP2′s molecular role. MECP2 duplication syndrome (MDS) is an overexpression in the same gene, and patients have substantial overlap in phenotype with RTT [10]. The pool of sequencing data for these diseases will only grow with time, and there is currently no centralized resource for it. Existing MeCP2-related disease databases are either primarily patient registries (Rett Database Network) [11], or they focus on mutation information, such as the IRSA North American Database [12] and RettBASE [13]. To fill this need, we created MECP2pedia, a database for molecular MeCP2. MECP2pedia is a uniformly processed and expansive collection of MeCP2 transcriptomic data, with readily accessible processed mouse data (expression, quality information, genomic tracks, and differential expression) that researchers can compare across any set of studies or data characteristics.

In this study, we demonstrate a comprehensive approach by curating a vast resource of transcriptomic data and then unbiasedly analyzing and interpreting the results to understand expression dysregulations. We derived a consensus common core of misregulated genes, which we delineated into consistently up- and downregulated. We validated this separation through unbiased clustering and discovered distinct functional characteristics between the up and the down cores. We further confirmed this robust core with enrichment across species and comparisons to other disease models. Finally, power analysis showed how high of a false-negative rate the average individual transcriptomic profile incurs through a low sample number, and the presence of strong batch effects in these data demonstrates another problematic hurdle for researchers.

Data integration is worthwhile but non-trivial. Batch effects, lack of power, reproducibility, and robustness are major hurdles for research. Big data helps mitigate these issues. Our approach to transcriptomic disease research yields results that are better and more complete than those attainable by conducting an individual experiment. Our demonstrated methodology can be applied broadly to other biological questions.

## 2. Results

### 2.1. Comprehensive Resource of MeCP2 Transcriptomes

The MECP2pedia portal can be accessed at http://www.mecp2pedia.org/. To generate this resource, we queried the National Center for Biotechnology Information (NCBI) Gene Expression Omnibus (GEO) [14] on 7 August 2019 for “MeCP2” and then filtered the results for “expression profiling by high throughput sequencing”. This search resulted in a preliminary list of 47 GEO entries. We filtered this list for entries dated 2015 or later and then further filtered for entries with at least two RNA-Seq wild-type samples and a treatment labeled either “knockout”, “Rett”, “point mutation”, “transgenic”, “overexpression”, or “MeCP2 duplication”. We retained 27 GEO entries, which we downloaded and processed with a uniform, streamlined Python pipeline (Figure 1A). In total, our processing yielded 546 sequence read archive (SRA) files, 753 FASTQ files, and 493 BAM (alignment) files. The total disk space used for processing was about 6 TB, and processing took about 2400 computing hours. The number of mouse samples per study is detailed in Appendix A. For each of the processed samples, we generated raw read quality, alignment quality, and track information. For each study, we collected meta information and data characteristics (Figure 1B), and then, we aggregated all samples with matching characteristics into “contrasts” (a comparison of the expression between two groups) for differential gene analysis. Mouse studies typically contained one to two contrasts, with the exception of four studies that, respectively, contained 3, 4, 6, and 10 contrasts (bar width in Figure 1B). The 27 GEO entries provided a total of 58 contrasts, with 43 from mouse studies, 10 from humans, and 5 from other species. As this work grew, to both stay up to date with data and diversify our sources, both ArrayExpress and European Nucleotide Archive (ENA) were queried on 13 September 2022 for “MeCP2”. Identical inclusion criteria were applied, and 6 mouse studies comprising 11 contrasts were retained, all from ENA. All analysis, core, and figures were made with only the initial GEO data. A list of mouse contrasts and their metadata is provided in Appendix A.

Portal users can quickly and easily compare across studies the expression of specific genes of interest (Appendix A). Queries can be carried out individually or as a multi-gene search. Bar and scatter plots are available for each gene to show significance and fold change, and these results can be filtered across the uniformly annotated data characteristics. TPM (Transcripts Per Kilobase Million) is shown to allow the comparison of expression between contrasts [15]. Users can browse genome tracks by individual study, and studies can be compared to one another. Metadata for each contrast, raw read quality, and alignment quality are all available for each study. A “significant genes” tab allows users to filter genes using an FDR (false-discovery rate) and log2 fold change thresholds for each contrast, and this information is also downloadable.

### 2.2. MeCP2 Transcriptomics in Mice Reveal a Common Core of Misregulated Genes

Our collected data are noteworthy for comprehensiveness and heterogeneity (Appendix A). Tissue, mutation, and cell fraction are all highly variable across the breadth of collected RNA-Seq, giving users the most complete picture possible of MeCP2′s transcriptomic role. To understand these changes per study, we examined fold change and FDR from the differential gene expression (DEG) analysis, and as shown in Figure 2A, the ratio of significantly upregulated to downregulated genes is generally similar across the 43 contrasts. This finding is consistent with MeCP2′s reported role in both gene activation and repression [16]. Furthermore, changes in the majority of the dysregulated genes are less than two-fold. This magnitude of change is low when compared to other mouse disease models, which have substantial quantities of genes with changes greater than two-fold [17].

In a traditional RNA-Seq study with one contrast (a comparison of the expression between two groups), a gene is considered a DEG based on FDR thresholding and/or fold change cutoffs. However, there are many contrasts in our data, and a truly biologically important DEG should be observed consistently in several contrasts. We examined the common FDR thresholds of 10% (0.1), 5% (0.05), and 1% (0.01), and due to MeCP2′s low magnitude of gene dysregulation, no fold change cutoff was used. We then examined number of contrast cutoffs of 1 (~2.5% of total contrasts), 4 (~10%), 12 (~30%), and 20 (~50%). When examining the FDR thresholds for a specific contrast cutoff (Figure 2B and Appendix A, per row), DEGs did not differ much. However, when comparing different numbers of contrasts for a specific FDR threshold (Figure 2B, per column), we observed large differences in DEG numbers. Moreover, when a gene is a DEG in many contrasts, it is likely to be either consistently upregulated or consistently downregulated. This indicates that the number of contrast threshold is a critical filter in establishing a robust set of DEGs. Thus, going forward, we set a strict FDR of 1% and a number of contrasts cutoff (10%) for common core DEGs, resulting in 2971 genes. Using average fold change, the common core DEGs were sorted into either core up (1666) or core down (1305) for further analysis. Mouse core DEG lists are in Appendix A.

We next investigated properties unique to the common core DEGs. First, we examined the annotations of these genes in relation to all the genes in the genome [18] and to all expressed genes. We observed that the common core DEGs have a notably high proportion of protein-coding genes. Specifically, while protein-coding genes constitute only 40.8% (21,922 out of 53,661) of all the genes and 57.6% (18,748 out of 32,551) of the expressed genes, they spike to 94% (2794 out of 2971) in the common core DEGs (Figure 2C). Next, due to MeCP2′s role in both chromatin structure [19] and various epigenetic features [20], we examined the common core DEGs for positional bias across the mouse genome. Within each chromosome, there are differences in positional distribution between the up and down core DEGs (Figure 2D and Appendix A). One striking example is on chromosome 8, where circular binary segmentation (CBS) [21] identified a stretch at the beginning of the chromosome (4,375,343–49,522,639) with many upregulated genes.

To further examine MeCP2′s regulatory role on these common core DEGs in an unbiased manner, we carried out unsupervised Leiden clustering [22]. From the nine clusters obtained, the two largest clusters (clusters 0 and 1) consisted mainly of the consensus core up and core down genes, respectively (Figure 3A). The clear directional separation of these clusters validates our core DEG selection methodology. Subsequent gene ontology (GO) analysis of these two clusters showed an enrichment in RNA Polymerase II (Pol2) and other transcription-related terms for the upregulated cluster (cluster 0) and an enrichment in neuronal and general nervous system-related terms for the downregulated cluster (cluster 1) (Figure 3B). Both up- and downregulated clusters displayed significant enrichment for cell differentiation, signal transduction, and general developmental terms. Our computational approach therefore provides some evidence not only the roles of MeCP2 as both activator and repressor but also established the core genes and functions involved.

We further examined the common core DEGs’ expression changes using a heatmap with annotation of contrast with cell fraction (Figure 3C). The unsupervised clustering shown on the heatmap categorizes the contrasts into three groups: (1) a mixture of all three types of cell fractions, (2) mainly with nucleus, and (3) mainly with whole cell. The common core DEGs are concordantly changed in about half of the contrasts, which fall into the first group of mixed-cell fractions. The two largest gene clusters identified from Leiden clustering are strongly up- and downregulated in this concordant set. Genes have lower expression changes in the second (nucleus) group. Notably, contrasts from the whole cell are categorized into two separate groups. The expression changes of common core DEGs are stronger and concordant in the first group and then weaker but still concordantly changed in the 17 contrasts of the third group. This is consistent with our findings in Appendix A, in which we observed solid overlap between common core DEGs and the common cores redefined separately by their sequenced cell fraction. Although the bulk of our data is comprised of whole cell, this picture of the transcriptome is not drastically different from MeCP2-dependent expression in the chromatin or the nucleus.

### 2.3. Cross-Species and Cross-Disease Comparisons of MeCP2′s Transcriptomic Signature

As we seek to understand the broad and complex role of MeCP2, mouse data alone re insufficient. Accordingly, we uniformly processed three human datasets, yielding seven contrasts. Human data yield fewer DEGs on average than mouse data, but the DEGs have similar ranges in fold change (Figure 4A) and a similar proportion of upregulation versus downregulation. The overlap between DEGs from these ten contrasts is low (Appendix A), suggesting limited homogeneity in the molecular signature of human MeCP2 dysregulation. This heterogeneity may reflect the fact that seven (GSE51607_1-4) out of the ten contrasts are from cell models, whereas the other three are from postmortem brain tissues. This separation is also seen in the heatmap of expression changes of human data on the mouse common core DEGs using unsupervised hierarchical clustering (Figure 4B). This heatmap shows limited qualitative correlation between the direction of human and mouse common core DEG dysregulation.

We also found transcriptome data for rat (*Rattus norvegicus*), macaque (*Macaca fascicularis*), and zebrafish (*Danio rerio*) MeCP2 models, which we processed and compared to the mouse common core. We found two rat studies with one contrast each, one monkey study with two contrasts, and one zebrafish study with one contrast. These data yield more DEGs than the human data, with roughly even proportions of up and down gene dysregulation. After associating DEGs to mouse orthologs, we found some overlap between these cross-species models and the mouse common core, with more overlap seen between mouse and rat than with other species (Appendix A). Figure 4B qualitatively shows a correlation between the mouse core and the rat data as well as some of the monkey data.

To provide a quantitative correlation between these data sets, we performed gene set enrichment analysis (GSEA). Pre-ranked analysis was run with our up and down mouse cores as gene sets (Figure 4C). Ranked lists from the 15 non-mouse contrasts were checked for overrepresentation in both up and down cores. We expect contrasts to be positively enriched in the up core and negatively enriched in the down core. The rat model has the best enrichment concordance, with both contrasts enriched as expected. Overall, these contrasts were more significantly negatively enriched in the down core than significantly positively enriched in the up core.

RTT and autism spectrum disorder (ASD) share a range of similar symptoms including loss of social, cognitive, and language skills. Altered MeCP2 expression is also commonly detected in autism brain samples [23]. Therefore, we hypothesized that our MeCP2 common core would display significant overlap with perturbed genes of other established ASD models. To generate an autism common core for comparison to our MeCP2 common core, we explored the expression changes in eight ASD models selected from the Simons Foundation Autism Research Initiative (SFARI) [24] (Figure 4D, Appendix A). All experiments involving knockdown or modification of the SFARI mouse model genes were retrieved from the ARCHS4 database. Across the eight target model genes, we processed 223 samples in 18 studies from 15 authors, from which we generated 28 contrasts. In our initial attempts to generate an overall ASD core, we found that expression changes were not generally concordant across contrasts (Appendix A), which was expected, as the contrasts contained a wide range of model genes and experimental procedures.

We thus analyzed the contrasts individually and performed Fisher’s exact test to determine the significance of overlap between each contrast and the MeCP2 core. We observed significant overlap in five contrasts (Figure 4E), representing five studies and four model genes (ADNP, ARID1B, CHD8, and SHANK3). We plotted the fold change of MeCP2 core genes in these five contrasts (Appendix A). We focused further on three of these contrasts (two CHD8 and one ADNP) with the largest DEG counts and most significant overlap and found that the significant overlap persisted even when considering only genes perturbed in the same direction in the contrast and the MeCP2 core (Figure 4F).

We then carried out GO analysis on the significantly overlapping gene sets. GO analysis of genes upregulated in both the MeCP2 core and a CHD8 contrast reveals significant enrichment for Pol2-arelated terms, while downregulated genes are enriched for nervous system development terms (Appendix A). This is consistent with our observations for up and down genes in the MeCP2 core. Even with the much smaller set of genes (~400 up and ~500 down in the overlap set compared to ~1600 up and ~1400 down in the MeCP2 core), we observed a similar gene ontology signal.

### 2.4. Sample Size Has a Major Impact on DEG Detection

MeCP2 interacts with other genes in an expansive manner [25], which lends our transcriptome data a low signal-to-noise ratio (SNR). This contributes to the limited number of clear consensus expression targets as well as the high degree of discordance in many individual data sets. To increase data detection sensitivity with low SNR, the number of samples therefore plays an important role. Yet, published MeCP2 studies often fail to meet to the conventional recommendation of at least six biological replicates [26]. To learn whether this problem limited the DEGs delineated from MeCP2 studies, we performed differential gene analysis on replicate down-sampled subsets of the data set with the highest number of replicates (GSE128178 Contrast 1). We found that the number of replicates has a negative correlation with the number of DEGs detected (Figure 5A). With no fold change cutoff, we could not saturate the number of detected DEGs with as many as ten replicates, which is far more sequencing than found in most published MeCP2 datasets. With a mild fold change cutoff (10% changed), the number of additionally detected DEGs in higher sample counts still did not appear close to saturation. Some saturation and flattening of the power curve began to occur with a 20% fold change cutoff.

To confirm our findings, we repeated the analysis using an RNA-Seq dataset in a psoriatic skin disease model (GSE63979) [27]. We chose this dataset for its high sample size and to understand how different transcriptomic SNRs affect the optimal number of replicates. We found relatively similar patterns (Figure 5B) but a reduced DEG loss effect from fold change thresholding. This finding confirms that the impact of replicate number on detected DEG is not unique to MeCP2 or to disease models with low SNRs.

Trends are also verified across common FDR thresholds, and Rand index is computed between full and down-sampled gene sets to understand how much the down-sampled results differ from the full results (Appendix A). Appendix A shows the differences between SNR in MeCP2 and psoriatic skin disease. The low power in MeCP2 data also supports our choice to require our common core DEGs to appear in just 10% of the contrasts.

To examine bias in and created by undetected DEGs in smaller n (sample number) studies, we first found the supersets of genes comprising each DEG cutoff number. For each DEG, we then computed average absolute log2 fold change (across contrasts). The higher n analyses detected DEGs at lower fold changes than analyses with lower sample numbers (Figure 5, last column), demonstrating that DEG sets from different sample sizes are affected differently by fold change cutoffs. A cutoff that removes many genes from a high sample size experiment may remove very few genes from a low sample size experiment. Moreover, the DEGs missed because they have too few biological replicates that are those with subtle perturbation, which is especially problematic for disease models with a low SNR. Researchers should be aware of this phenomenon when choosing fold change cutoffs and evaluating results.

### 2.5. Batch and Technical Variation Must Be Overcome in Order to Integrate and Understand Data

Relevant non-biological factors, commonly called batch factors, are often present in a given researcher’s data. These batch factors could reflect the use of particular tissues, mouse litters, sequencing platforms, or other variables. Therefore, it is important to integrate and analyze transcriptomic data across years of work with dozens of meta-characteristics. We saw extreme batch effects on the raw count values for all samples included in MECP2pedia in that the samples were initially segregated by study (Figure 6A). After batch correction and normalizing the raw counts, the segregation was reduced, but samples remained grouped by study. This finding indicates that batch correction and normalization failed to fully resolve this batch effect. When we examined all available meta-characteristics, we observed that the clustering weakly overlapped with the studies’ prominent meta-characteristics, such as cell fraction, tissue, and gender (Appendix A). As a basis for comparison, we performed the same analysis on a set of 316 samples from nine neurological degeneration studies analyzed in Wan et al. [28]. We observed a similar outcome: an extreme batch effect on raw data and an inability of batch effect correction and normalization algorithms to fully remove this batch effect (Figure 6A and Appendix A).

To better understand the concordance between sexes in MeCP2 models, we compared their molecular signatures. RTT occurs almost exclusively in females, but most MeCP2 studies are carried out in male mice due to their relative ease of use and availability [29]. Hence, we had only one fair comparison between male and female models of similar age and tissue (Figure 6B). We found that the DEGs from the male mouse model had no overlap with the female model at one time point and minimal overlap at a second time point. Furthermore, the direction of dysregulated genes did not show a concordant trend. More data are needed to understand these differences, but researchers should be mindful of sex in their experimental design, especially in RTT.

## 3. Discussion

With the low cost and high quality of modern sequencing, the scope of publicly available data is rapidly expanding. However, to make the leap from big data to big insights, curation and automation are essential. In addition to gleaning insights from the portal’s data, researchers can compare their own novel data to this convenient aggregation of publicly available transcriptome profiles. We plan to add new datasets to the portal and also add a feature allowing users to upload their own processed data for comparison.

MeCP2 transcriptomics from published studies often seem contradictory, perhaps due to low SNR and disparities in experimental design. However, by bringing a robust approach to the integration of big data, we uncovered a common core of MeCP2 DEGs with high concordance across studies, suggesting that MeCP2′s core function is universal across the examined breadth of tissues, cell fractions, mutations, and mouse strains. The positional bias in common core distribution demonstrates MeCP2′s importance to the epigenome, while unbiased clustering further underscores the concordance in core genes, providing insight into their regulatory relationship. When the clusters were enriched to GO terms associated with Pol2 activation and neuronal function, respectively, the two main clusters from the unsupervised clustering correspond to up- and downregulated genes. Exploration of the smaller mixed clusters may similarly reveal insights into other proposed mechanisms of MeCP2 action [30,31].

Examination of diverse MeCP2 models provides quantitative comparisons of MeCP2′s dysregulatory molecular signature across species. Robust enrichment across species in the downregulated DEGs supports the core we have derived, and this finding can be further explored in the context of our delineation on up versus down core genes. Comparison across disease models revealed links between MeCP2 disorders and common ASD models. Specifically, we observed highly significant concordant overlap between genes perturbed in the MeCP2 core and two ASD models: ADNP and CHD8. The RNA Pol2 function in the upregulated genes and neuronal development in the downregulated genes we observed in the MeCP2 core are also enriched genes common to MeCP2, ADNP, and CHD8 models. These overlaps could provide the basis for deeper exploration of the relationship between MeCP2 disorders and other autistic spectrum disorders.

Since small sample size leads to low statistical power, the validity, specificity, and robustness of the DEGs delineated from a single study may be unreliable [26]. We validated this concern in our analysis of multiple datasets, and the problem is exacerbated when the SNR is low. Researchers who are interested in perturbations with a small effect should therefore aim to generate large datasets, with 10 or more samples per condition. If resources are limited, six samples per condition would be a good compromise. However, most studies failed to meet even this lower threshold, which may explain the lack of consensus conclusions across independent studies and their failure to capture the complete picture of transcriptomic perturbation. Another limitation of this resource is the low availability of female data, especially given Rett’s primary impact on girls.

Our expansive study sheds light on the high variability in transcriptomic profiles of a disease model across different tissues, ages, sexes, species, and other biological and technical artifacts. The specific experimental conditions of a single study therefore cannot capture the complete picture of transcriptomic changes. Individual researchers will always be limited in the data that they can personally generate. Our big data integration platform solves this problem, making it invaluable for scientists studying complex diseases such as RTT and MDS.

## 4. Materials and Methods

### 4.1. Data Collection

To generate this resource, we queried NCBI GEO (https://www.ncbi.nlm.nih.gov/gds) on 7 August 2019 for “MeCP2” and then filtered the results for “expression profiling by high throughput sequencing”. This search resulted in a preliminary list of 47 GEO entries. We filtered this list for entries dated 2015 or later (with an exception made for GSE51607 due to sparsity of human data) and then further filtered for entries with at least two RNA-Seq wild-type samples each and a treatment labeled either “Knock out”, “Rett”, “Point Mutation”, “Transgenic”, “Overexpression”, or “MeCP2 Duplication”. We retained 26 GEO entries. Datasets with no associated publications were included. We subsequently added three more studies, namely GSE123941, GSE128178, and GSE123372, based on the scope and relevance of their data. All preprocessing was carried out with a uniform, streamlined Python pipeline. SRA files were downloaded with prefetch from SRAtoolkit.2.9.6-1-centos_linux64 [32] and then converted to fastq with fasterq-dump version 2.3.5, using the --split-files option. Fastq files were then checked for quality with FastQC version 0.11.7 [33]. Both ArrayExpress and European Nucleotide Archive (ENA) were queried on 13 September 2022 for “MeCP2”. Identical inclusion criteria were applied, and six mouse studies comprising 11 contrasts were retained, all from ENA. These were downloaded with wget and processed identically to the GEO studies.

### 4.2. Mouse Data Processing

Mouse samples were aligned to GENCODE GRCm38p6 primary assembly, version 18 (https://www.gencodegenes.org/mouse/release_M18.html, accessed 18 July 2019), with STAR version 2.6.0a [34] at default parameters. STAR gene quantifications were used (--quantMode GeneCounts). The assembly also contained an appended copy of human MeCP2 from hg38. BigWig files were generated with bamCoverage version 3.3.1 from deepTools [35]. We assessed alignment quality with RSeQC geneBody_coverage and read_distribution, both version 3.0.0. Overall quality per study was examined with MultiQC v1.7 [36]. DEG analysis was performed in R version 3.5.2 (Eggshell Igloo) with DESeq2 version 1.24.0 [37] after loose expression filtering (per contrast, a gene must have a sum of 10 counts in at least half the samples).

Data were not trimmed except for samples SRR3679844, SRR3679845, SRR3679848, SRR3679849, SRR3679852, and SRR3679853 from GSE83474 due to slight anomalies in their raw sequences. The trim was performed with Trimmomatic-0.36 [38] using the following parameters: PE ILLUMINACLIP:TruSeq3-PE.fa:2:30:10 HEADCROP:8

### 4.3. Data Annotation

We downloaded SRA run tables for each GEO entry. Data characteristics of interest were: genotype, organism, experiment, run, sample name, cellular fraction, strain, age, cell line, cell type, tissue, sex, mutation, and disease. Incomplete run tables were filled in from the contents of their publications, if available. After processing, we annotated samples for sequencing depth and contrasts for number of DEG at FDR < 0.01 with no fold change cutoff.

### 4.4. Data Visualization

Unless otherwise specified, plots were made in R with ggplot2 version 3.2.1 [39]. Box plot elements are as follows: minimum, first quartile, median, third quartile, and maximum.

### 4.5. Portal Development

Mouse data are in the portal, while human and cross species are not. Python pipeline analysis results and GEO sample information were parsed and saved using the MongoDB NoSQL database. The web server was written in JavaScript and serves an API that gives access to the data and the web portal application. Data visualization uses the D3.js library and IGV.js [40] for genomics tracks. ENA studies are in the portal but not included in the analysis and common core.

### 4.6. Core Gene Identification and Clustering

First, fold changes with FDR > 0.01 were set to zero. Contrasts with all non-significant fold changes were removed. To generate our set of core genes, we kept only genes with non-zero fold change in four or more contrasts (10%). This resulted in a set of 2971 genes for further analysis. For consistency in the analysis and the visualization of gene regulation direction, we inverted the direction of fold change for the four contrasts of the MDS model (GSE123372_3, GSE66870_2, GSE71235_1, GSE71235_2).

We also identified alternative core genes based on contrast metadata characteristics. For each type of cell fraction (chromatin, nucleus, whole cell) and the cortex and forebrain tissue types, we performed the same core gene identification filtering as we did for the main core. We considered only the contrasts with the metadata feature of interest and kept only genes with significant and non-zero fold change in at least 10% of the contrasts. We included an upset plot generated with the R package UpSetR [41] as Appendix A.

After identifying the set of core genes, we assigned unsupervised clusters using the Scanpy [42] implementation of the Leiden algorithm with the parameters: number of neighbors = 45 and resolution = 0.5. Then, we generated UMAP coordinates with the parameters: number of neighbors = 45, minimum distance = 0.1, and spread = 10. We then used UMAP’s Scanpy implementation to generate plots of the unsupervised clusters as well as up- and downregulation.

### 4.7. Core Gene Characteristics and Location

We annotated mouse genes with GRCm38p6 primary, version 18 from GENCODE. Genes were sorted into eight super-categories to show broad function. Expressed genes (32,539) is a superset of the genes that pass the expression filter in any contrast.

Core genes were plotted by their TSS (Transcription start site). Chromosome 8 was plotted with an equivalent number of randomly drawn non-core genes (150) to show the strength of its regional core up DEG enrichment. We validated this trend with the CBS algorithm implemented with R Package PSCBS version 0.65.0. The core up, core down, and non-core genes were, respectively, assigned values of 6, 0, and 3 for segmentation detection and plotting.

### 4.8. GO Analysis

GO analysis was performed using the Python GOAtools package [43]. We performed an enrichment analysis on each MeCP2 Leiden cluster, using all NCBI protein coding mouse genes as the background set. For each Leiden cluster, we retained the top six biological process GO terms by frequency. We used the same methodology to perform GO enrichment on overlapping MeCP2 and ASD core genes, also retaining the top six biological process GO terms by frequency.

### 4.9. Human Data Processing and Comparative Analysis

Human data were aligned to GRCh38p12 primary assembly, version 28 from GENCODE with STAR. BigWig generation, assembly quality metrics, and DEG analysis were performed identically on mouse data. Human genes were then queried for their orthology to mouse genes with DIOPT 8.0 [44] using the “return only best match” option. Upset plots were made with function upset from Package UpSetR, version 1.4.0. Human and other metadata are available in Appendix A.

### 4.10. Other Model Data Processing and Comparative Analysis

Rat data were aligned to the Rnor_6.0 toplevel assembly and annotated with Rnor_6.0.99 from Ensembl [45]. Zebrafish data were aligned to Danio_rerio.GRCz11 toplevel assembly and annotated with Danio_rerio.GRCz11.100 from Ensembl. The orthology tables for rat and zebrafish genes were retrieved from DIOPT, as was done for human data. Macaque data were aligned to Macaca_fascicularis_5.0 and annotated with Macaca_fascicularis_5.0.100 from Ensembl. Macaque gene orthology data were retrieved with the function getBM from R package biomaRt, version 2.38.0 [46]. The only data trimmed were GSE57974, using Trimmomatic-0.36 with the following parameters: LEADING:3 TRAILING:20 MINLEN:50. Genotype labels for GSE87855 were inferred based on MeCP2 level.

Heatmaps were made in R with pheatmap package version 1.0.12 [47] using log2 fold change, no clustering on rows, and clustering_distance_cols = “euclidean”. For consistency in the analysis and the visualization of gene regulation direction, we inverted the direction of log2 fold change for the MDS contrast (GSE57974_1)

### 4.11. GSEA

GSEA [48,49] version 4.1.0 Pre-Ranked was run with default parameters besides set_max of 100,000 and set_min of 1. Ranking values were computed per gene as –log10(adjusted *p* value) * log2 fold change. For consistency in the analysis and the visualization of gene regulation direction, we inverted the direction of normalized enrichment score fold change for the MDS contrast (GSE57974_1). GSEA results are available as Appendix A.

### 4.12. ASD Model Comparison

All experiments involving knockdown or modification of the SFARI mouse model genes were retrieved from the ARCHS4 database. There were 18 such studies, which we processed using DESeq2 to generate DEGs across 28 contrasts. We retained only DEGs with FDR < 0.01. With the DEGs for each ASD contrast, we used the hypergeometric and Fisher’s exact tests to determine the significance of overlap with the set of all the MeCP2 core genes and also both up- and downregulated subsets. Computed ASD contrast fold change is available as Appendix A.

We also used the previously described process to perform GO analysis on genes in the intersection of the ASD contrasts and MeCP2 cores.

### 4.13. Down Sampling Analysis

MeCP2 data are from GSE128178. All 10 samples of wild-type and knockout whole-cell data were randomly selected to create 100 random drawings at each different sample number. (For instance, sample 9 was one random sample removed from each genotype, and so on.) Once selected, samples were normalized with each other and analyzed with the same DEG methodology detailed above. The Rand index was then computed using the rand.index function in R (fossil, version 0.4.0) [50]. Vectors indicating if each gene was a DEG for a particular run were compared to the vectors of DEGs for the complete contrast of 10 wild-type and 10 knockout samples, respectively.

Psoriatic skin data are from GSE63979 (SRP050971). This study contains the total RNA-Seq data of nine normal skin samples, nine lesional psoriatic samples, and twenty-seven uninvolved psoriatic samples. In order to conduct the downsampling analysis between two groups with the same sample number, only normal skin samples and lesional psoriatic samples were chosen. For each phenotype, all nine samples were randomly selected to create 100 random drawings at each different sample number. For instance, sample 8 was one random sample removed from each phenotype, and so on. The DEG analysis and Rand index comparison on psoriatic skin data was the same used for MeCP2 data.

### 4.14. Technical Variation/Batch Effect Analysis

All raw MeCP2 mouse expression value data were dimensionally reduced using UMAP version 0.2.6.0 in R and plotted with color for contrast of origin. ComBat_seq [51] from R package sva, version 3.36.0, was then run with contrast as batch to deconvolute the data. ComBat_seq-normalized data were then size factor-normalized with DESeq2 and plotted again.

To provide an alternate dataset to evaluate batch effect, we used a set of Alzheimer’s disease data stored on Synapse from Wan et al. [28]. We retrieved raw count data and plotted the samples using UMAP. Count data were normalized with DESeq2, and then, we used the ComBat_seq function to attempt to control for batch effect.

### 4.15. Sex Comparison

The compared contrasts are GSE90736_1, GSE90736_2, and GSE66211_1. Genes were considered DEGs if they passed FDR < 0.01. The plot was made with R package VennDiagram version 1.6.20.

## Figures and Tables

**Figure 1 ijms-24-05122-f001:**
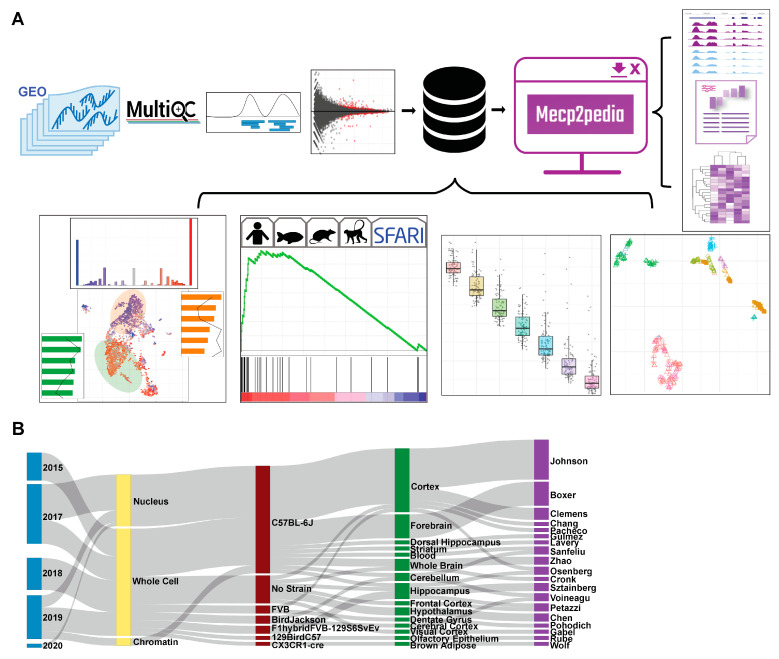
Overview of data and workflow. (**A**) Workflow for portal data and analysis. Processing is uniform and unbiased. Quality, track, and DEG analysis results are available in an intuitive and comparable manner through our portal. (**B**) Sankey plot on major characteristics per contrast of the collected mouse data (date, cell fraction, strain, tissue, first author). Metadata were collected with sequence data and then standardized.

**Figure 2 ijms-24-05122-f002:**
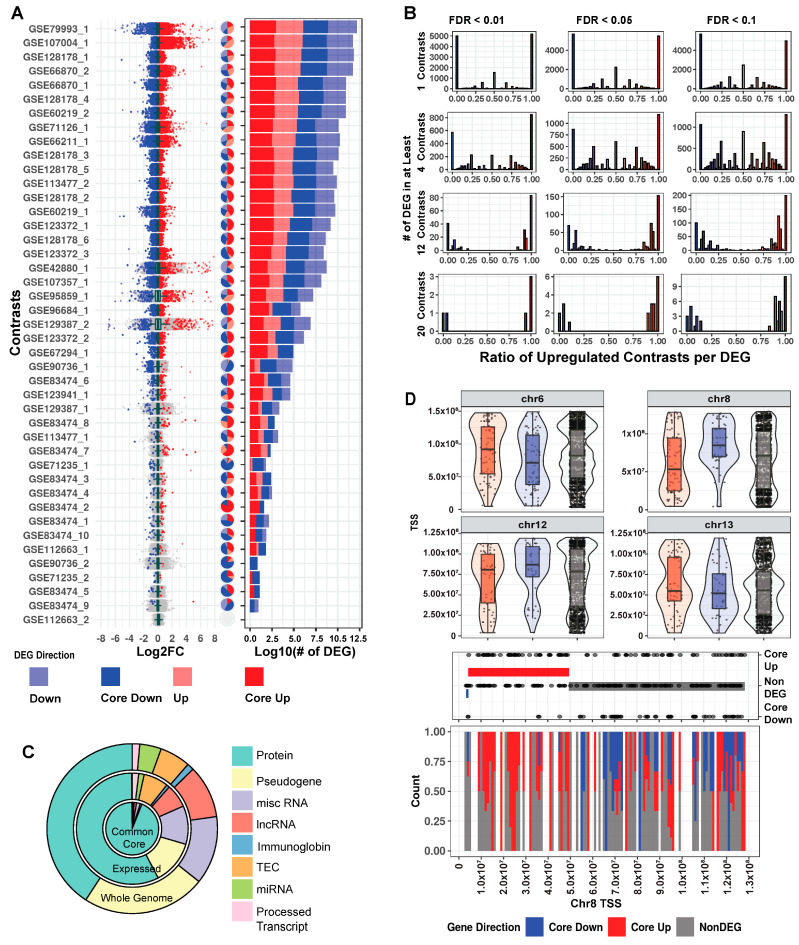
Mouse transcriptome common core. (**A**) Distribution of log2 fold change across contrasts with significant (FDR < 0.01) DEGs. Dark blue and dark red, respectively, indicate genes that are core down and core up, and pale blue/pale red, respectively, indicate down and up DEG. Pie charts with the same annotation colors show what percentage of each contrast’s DEGs falls into each category. Stacked bar charts with the same annotation colors show each contrast’s DEG quantity. (**B**) Histograms of significantly up and downregulated genes cut for different FDR thresholds and the number of total contrasts in which a DEG appears. Genes at the extreme ratios of 0 or 1 percent upregulated are highly concordant across contrasts, whereas genes that fall into the middle are discordant. For consistency in this analysis, we inverted the direction of fold change for the four contrasts of the TG model. (**C**). Genes as annotated by Gencode and condensed into eight broad categories. We considered 53,661 genes from our annotation and found 32,539 that passed our expression filter in at least one contrast. The common core (FDR < 0.01 in at least four contrasts) is comprised of 2971 genes. (**D**). Exploration of genome location trends in the common core. All non-DEGs are plotted in the upper portion of the panel, and violin plots show areas of gene density. Chromosome 8 was selected for further examination in the lower half of the panel, with baseline genes in equal quantity to the core genes (150) also plotted. CBS method is used to identify trends in the up/down/baseline genes. The bands on middle dot plot show these CBS results, and the lower stack plot shows the density of up, down, and baseline genes on chromosome 8.

**Figure 3 ijms-24-05122-f003:**
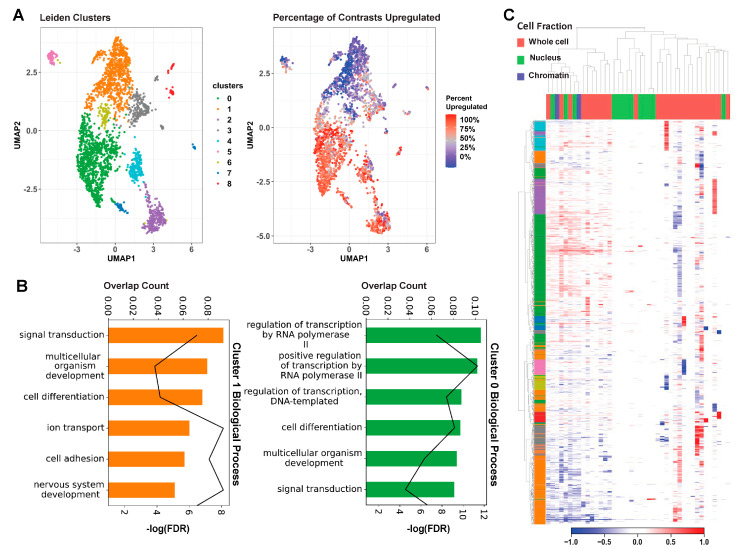
Unsupervised clustering on core genes. (**A**). The left UMAP plot colors each gene by cluster, assigned through unsupervised Leiden clustering. The right UMAP plot displays the percentage of contrasts in which the gene was upregulated on a spectrum of red (upregulated in all contrasts) to blue (downregulated in all contrasts). We can see that the green and orange clusters roughly encompass the up and downregulated genes. (**B**). Results of GO analysis on Leiden clusters 0 and 1. The bar colors correspond to cluster. Bar length represents the proportion of genes enriched with the term in the cluster, and the line plot represents the FDR of the enrichment. (**C**) Heatmap of contrasts (columns) by genes (rows). Contrasts are labelled based on the experiment’s cell fraction, and genes are labelled based on their Leiden cluster. We can see the general downregulation in the orange cluster and the upregulation in the green cluster. We can also see from this figure that the other clusters are generally caused by extreme deviations in one or two studies.

**Figure 4 ijms-24-05122-f004:**
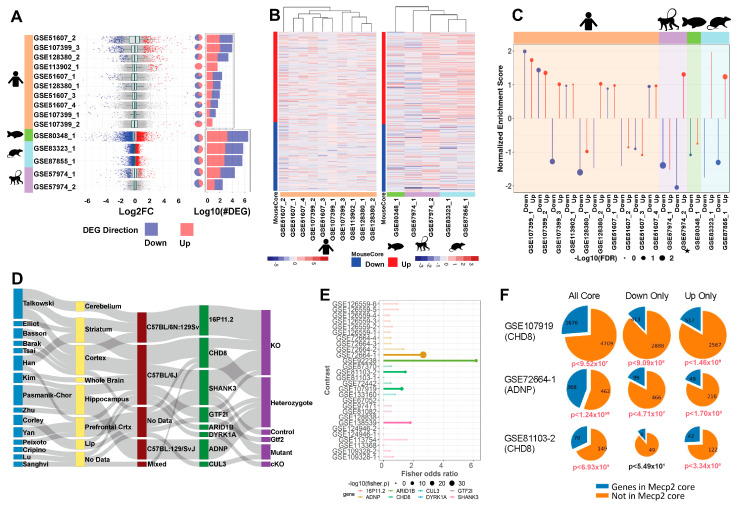
Mouse transcriptome translation to other models. (**A**). Distribution of log2 fold change across contrasts with significant (FDR < 0.01) DEGs. Blue and red, respectively, indicate down and up DEGs. Pie charts with the same annotation coloring show the percentage of each contrast’s DEGs in each category. Stacked bar charts with the same annotation color show each contrasts’ DEG quantity. The upper 7 contrasts are human data, and the lower 5 are other species. (**B**). Heatmaps of log2 fold change plotted to compare direction of dysregulation to the consensus from mouse data. Genes examined are the mouse common core, and plots are annotated on mouse core down and mouse core up. (**C**). Per-contrast visualization of GSEA normalized enrichment score and FDR. Direction and color of line represents normalized enrichment score, and point size represents log10(FDR). Contrasts are grouped and shaded corresponding to their model of origin. MDS model is annotated with a small star. (**D**). Sankey plot of ASD contrast metadata characteristics. From left to right: first author, tissue, strain, target gene, and experimental procedure. (**E**). Fisher’s exact test results. Points sized by –log10(*p*-value), length determined by odds ratio, data colored by gene. Points are opaque, and overlap to MeCP2 core is considered significant if the Fisher *p*-value is less than 0.05. (**F**). Pie charts show the magnitude of overlap between selected ASD contrasts and the MeCP2 common core. Down and up only show genes changed in the same direction in both sets. *p*-values beneath each plot show the Fisher’s exact test significance of the overlap for each intersection, colored red if the *p*-value is less than 0.05.

**Figure 5 ijms-24-05122-f005:**
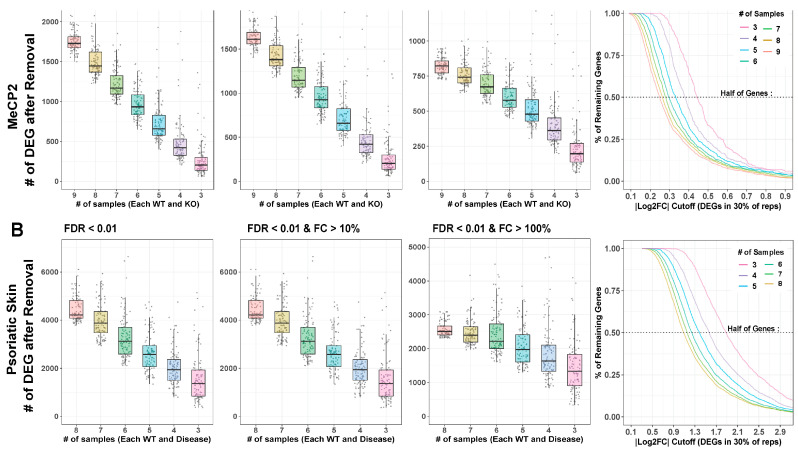
Downsampling analysis. (**A**,**B**). Large contrasts from MeCP2 and lesional psoriatic skin data were downsampled to smaller and more common experimental sample number, and DEG analysis was run on these subsets. Box plot and jitter points are plotted for resultant DEG numbers under each condition. Cutoffs for MeCP2 are sample numbers 9 through 3. Cutoffs for psoriatic data are sample numbers 8 through 3. Each cutoff number was repeated 100 times, with random samples discarded each time. Results are plotted at FDR < 0.01. MeCP2 data are fold change (FC) cutoff at any FC, FC > 10%, and FC > 20%. Psoriatic skin data are FC cutoff at any FC, FC > 10%, and FC > 200%. Curves indicate the percent of DEGs remaining at continuous |log2 fold change| cutoffs. Horizontal line indicates 50% of genes removed.

**Figure 6 ijms-24-05122-f006:**
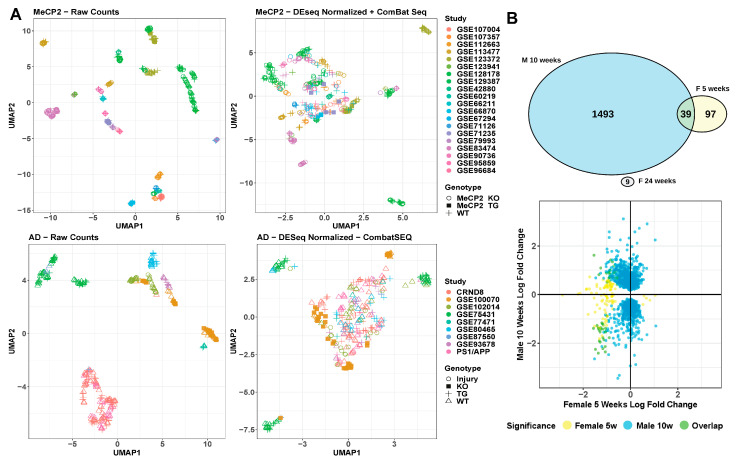
Technical variation/batch effect analysis. (**A**) UMAP visualization of raw and corrected data from MeCP2 and AD. (**B**). Comparison of cell type-matched (microglia), male-to-female contrasts by DEG overlap, and direction of misregulation. DEG are FDR < 0.01 and any fold change.

## Data Availability

All datasets are available from GEO under their specified accession numbers. Accession numbers are in figures, text, portal, and supplements. Processed mouse results are available through our portal, and other processed results are available as supplements. All code packages used and their versions are detailed in the Section 4. Processing pipeline is available here https://github.com/LiuzLab/mecp2sca1. All other code used to produce the findings of this study is available from the corresponding author upon request.

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
