# Peer review of "A Comprehensive and Integrative Approach to MeCP2 Disease Transcriptomics"

_ijms, 2023, doi:10.3390/ijms24065122_

Round 1

Reviewer 1 Report

Here Trostle et al combine large scale transcriptomic datasets across research groups/species/models with the goal of providing a more comprehensive analysis of the impact of MeCP2 mutations on gene dysregulation. Given the potential importance of this epigenetic regulator to both Rett syndrome and autism spectrum disorder, this dataset will be of great interest in the field of neurodevelopmental disorders. This study by Trostle et al is an excellent example of the potential utility of data aggregation and curation in understanding the pathogenesis of complex diseases. In general, I find that the data included and the resources described will be of interest to the field and merit publication. I have a few suggestions below for improvement of the manuscript to aide in readability and accessibility of the results to the field.

Major comments:

1)     The abstract is quite generic and does not provide a comprehensive description of the analytical methodology and key results. In particular, the final sentences seem out of place. The authors should provide here 1) details on their approach, 2) key findings including the gene sets identified across studies/organisms/models, 3) the benefits of this kind of meta-analytic/informatics-based approach.

2)     When describing the MeCp2pedia, it might be helpful to include, possibly as a supplement, examples of how the user interface appears coupled to the description on lines 111-120.

3)     The repeated use of the term “contrast” is somewhat confusing. This term could be better defined.

4)     In looking through the meta-data of the samples included (Table S1) there is a clear skewing toward datasets isolated from males. Given that Rett syndrome primarily impacts girls and rarely boys, this should be noted in the discussion as a limitation of the dataset.

5)     It would be useful, for examples where the authors look at overlap across MeCP2 samples or between species/models to isolate examples of genes with the highest overlap and note whether these make sense in the context of disease pathology or have been identified as causative in other studies. This would provide additional validation of this approach as a means to better utilize these high throughput methodologies in identification of novel mechanisms or druggable targets.

Author Response

1)     We appreciate the reviewers’ thoughtful comments, abstract has been overhauled to reflect these three points.

2)     Screenshots are added as supplemental figure 7 – figure legend as follows :

Supplemental Figure 7 : Portal demonstration

  1. Bar plot to compare one gene (Sdk1) across studies. Sdk1 is the top core gene by number of significant contrasts, and expression follows a clear general trend of up in Rett models, down in Mecp2 duplication syndrome models. Built in links to MARRVEL and Ensembl make follow up literature searches and biological understanding a seamless process. This can be filtered by tissue(s) and disease model of interest.
  2. Example heatmap result from multi-gene search using the top 3 core genes, Sdk1, Fgf11, and Cacna1g.
  3. TPM plot example showing a cross study expression comparison of Sdk1. This can be filtered by tissue(s) of interest.
  4. IGV genomic tracks at example zoomed to Mecp2 showing a 3v3 knockout data set. Any 1 or 2 studies can be loaded at the same time to compare.
  5. Significant genes tab accessible from the top panel of the portal, all summarized DESeq results are downloadable as a table. The data is filterable on p-adjusted and log2 fold change. The default gene order is based on the same criteria as our core genes – number of significant contrasts defined by p-adjusted < 0.01. Therefore, our top individual gene findings are conveniently presented here.

3)     This term was defined on line 153, but we will add a definition (a statistical comparison of the expression between two groups), the first time the term appears on line 98.

4)     Good point, we examine and comment some on this with figure 6B, but have added a sentence to discussion on line 417.

5)     Our core genes are the most consistent across the mouse data, figure 3 seeks to explore and validate them, particularly panel B with GO analysis. To look at our top core genes, users can click the “Significant genes” tab of the portal along the top bar. This shows how many mouse contrasts a given gene is significant in, with the top three genes being Sdk1, Fgf11, and Cacna1g. The first two lack phenotypic associations from OMIM, but Cacna1g in human has association to Spinocerebellar ataxia 42; also a neurological disorder. These sorts of follow up literature checks are facilitated by the built in links to MARRVEL and Ensembl in the web portal per gene.

The original purpose of our cross species analysis was to further validate the core genes, but technical noise (post-mortem tissue or cell culture) in human likely prevents us from seeing much overlap. Encouragingly, we see the best congruence to rat data, which makes sense from mouse species similarity.

To clarify this as well as address your 2nd point, example screenshots of the portal are added as supplemental figure 7.

Reviewer 2 Report

The manuscript by Trostle et al. describes a comprehensive approach of integration of public big data with the identification of a common perturbed core of genes involved in MeCP2 disease. The advantage of this approach is the possible application to any biological questions thanks to the complexity of the process analysis. The authors should include the following paper: RNA sequencing and proteomics approaches reveal novel deficits in the cortex of Mecp2-deficient mice, a model for Rett syndrome By Pacheco et al 2017.

Author Response

Thank you for the comments. The RNASeq from Pacheco et al is included as GSE96684, with the differential expression analysis accessible through the portal at http://www.mecp2pedia.org/result/contrast/GSE96684_1/deseq

Reviewer 3 Report

Dear Authors, the research article “A Comprehensive and Integrative Approach to MeCP2 Disease Transcriptomics” is well organized and report a very interesting study. You present MeCP2 transcriptome data from mouse model on public web database. The MECP2pedia portal is user friendly, complete, detailed, and useful. You did not conclude with clarification on the MeCP2’s molecular role but shed the light on the high the variability in transcriptomic profiles of MeCP2-models and across different tissues, ages, sexes, species, and other biological and technical artifacts, ad this is an interesting topic of discussion which has a broad spectrum and involves transcriptomic data related to different pathologies. The article is suitable for the publication with minor revision. I ask you to better highlight that MECP2pedia portal is based only on mouse models in the abstract and introduction.

Author Response

We appreciate the helpful comment. We have clarified this in the abstract and on  line 63.